# Quetiapine Albumin Nanoparticles as an Efficacious Platform for Brain Deposition and Potentially Improved Antipsychotic Activity

**DOI:** 10.3390/pharmaceutics15071785

**Published:** 2023-06-21

**Authors:** Hend Mohamed Abdel-Bar, Alaa S. Tulbah, Hany W. Darwish, Rania Salama, Ibrahim A. Naguib, Heba A. Yassin, Hadel A. Abo El-Enin

**Affiliations:** 1Department of Pharmaceutics, Faculty of Pharmacy, University of Sadat City, Sadat City 32897, Egypt; 2Institute of Pharmaceutical Science, Faculty of Life Sciences & Medicine, King’s College London, London SE1 9NH, UK; 3Department of Pharmaceutics, College of Pharmacy, Umm Al-Qura University, Makkah 21955, Saudi Arabia; astulbah@uqu.edu.sa; 4Department of Pharmaceutical Chemistry, College of Pharmacy, King Saud University, Riyadh 11451, Saudi Arabia; 5Macquarie Medical School, Faculty of Medicine, Health and Human Sciences, Macquarie University, Sydney, NSW 2109, Australia; rania.salama@mq.edu.au; 6Woolcock Institute of Medical Research, Glebe, NSW 2037, Australia; 7Department of Pharmaceutical Chemistry, College of Pharmacy, Taif University, Taif 21944, Saudi Arabia; i.abdelaal@tu.edu.sa; 8Department of Pharmaceutics, Faculty of Pharmacy, Badr University in Cairo (BUC), Badr City 11829, Egypt; heba_ahmed@buc.edu.eg; 9Department of Pharmaceutics, National Organization of Drug Control and Research (NODCAR) (Previously), Egyptian Drug Authority (Currently), Giza 12511, Egypt; hadelaboenin@outlook.com

**Keywords:** quetiapine, human serum albumin, nanoparticles, brain targeting, antipsychotic activity

## Abstract

Quetiapine (QP) is a second-generation short-acting antipsychotic drug extensively metabolized in the liver, producing pharmacologically inactive metabolites and leading to diminished bioavailability. Therefore, this study aimed to develop an intravenous QP albumin nanoparticles (NPs) system for improving QP antipsychotic activity and brain targeting. QP-loaded albumin NPs were prepared by the desolvation method. The fabricated NPs were characterized in terms of particle size, zeta potential, entrapment efficiency (EE%), and in vitro drug release. In vivo pharmacokinetics and biodistribution in rats were studied. In addition, the antipsychotic activity of the optimized platform was also investigated. Human serum albumin (HSA) concentration, pH, and stirring time were modulated to optimize QP albumin NPs with a particle size of 103.54 ± 2.36 nm and a QP EE% of 96.32 ± 3.98%. In addition, the intravenous administration of QP albumin NPs facilitated QP brain targeting with a 4.9-fold increase in targeting efficiency compared to the oral QP solution. The QP albumin NPs improved the QP antipsychotic activity, indicated by suppressing rats’ hypermobility and reducing the QP’s extrapyramidal side effects. The obtained results proposed that intravenous QP- NPs could improve QP brain targeting and its antipsychotic efficiency.

## 1. Introduction

Antipsychotic drugs have recently gained great attention due to increased usage worldwide, related to the high incidence of psychiatric disorders [1]. Quetiapine (QP) is atypical short-acting antipsychotic drug belonging to the dibenzothiazepine derivatives class [2], which is used in treating adults and adolescents with major depression, schizophrenia, and bipolar disorder attacks [2,3,4].

Oral QP shows extensive hepatic first-pass metabolism by the CYP3A4 enzyme, leading to low oral bioavailability (~9%) [5,6]. QP’s low oral bioavailability requires an increase in the dosage amount and its frequency to achieve the required therapeutic plasma levels. This affects the patient’s lifestyle and triggers dose-related side effects [5,7]. To overcome these limitations, platforms that directly deliver drugs to the brain are crucial [8].

The formulation of brain-targeting delivery systems is still considered a challenge due to the unique structure of the blood–brain barrier (BBB), which restricts the systemic delivery of the majority of neurotherapeutics. Albumin-based nanoparticles have unique features [9,10], including biocompatibility, biodegradability, high drug-binding capacity, long systemic circulation, and adequate cellular uptake [9,10,11]. Furthermore, functional groups of the albumin molecule, i.e., amino and carboxylic groups, can be utilized for surface modifications, enabling specific cellular targeting and avoiding the adverse drugs’ toxicity [12,13,14]. Albumin-carrier properties that improve its binding with several drugs are commonly reversible due to weak interactions such as hydrogen bonding, hydrophobic forces, ionic interactions, and van der Waal’s interactions [15,16].

The current study aims to develop human serum albumin (HSA) NPs of QP. The novelty of this work is the formulated QP albumin in nano-sized particles. Nanoparticles are effective in improving the stability of plasma protein hybrids with their unique properties in improving drug absorption and distribution, and facilitating brain targeting. The developed QP albumin NPs (QP-NP) were prepared using the desolvation method. The QP-NPs were statistically optimized using Box–Behnken design (BBD) to obtain a suitable platform with minimum particle sizes and maximum drug entrapment efficiencies. The in vivo pharmacokinetics and organ biodistribution, as well as the antipsychotic activity of the proposed system, were also investigated.

## 2. Materials and Methods

### 2.1. Materials

Quetiapine fumarate (QP), human serum albumin (HSA), and fetal calf serum (FCS) were purchased from Sigma Aldrich (St. Louis, MO, USA). Acetonitrile (HPLC grade), ethanol absolute (HPLC grade), Potassium dihydrogen orthophosphate, sodium hydroxide, and Tween 80 (polysorbate 80) were obtained from Fluka Chemika-BioChemika (Buchs, Switzerland). All other chemicals were of an analytical grade and were purchased from El-Nasr Pharmaceutical Chemicals Co. (Cairo, Egypt).

### 2.2. Formulation and Evaluation of Quetiapine HSA Nanoparticles

#### 2.2.1. Experimental Design

To prepare QP-NPs with minimum particle size and maximum entrapment efficiency (EE%), the BBD experimental design was used to choose the optimum experimental conditions. HSA concentration (mg/mL), pH, and stirring time (h) effects were studied (Table 1). The chosen factors and their corresponding levels were selected based on preliminary experimental results. The regression model was used to plot the 3D response surface plots for particle size (Y1) and EE% (Y2). The analysis of variance was used to assess the significance of the variables and interactions (ANOVA), where a *p*-value less than 0.05 was considered significant.

#### 2.2.2. Preparation of Quetiapine- HSA Nanoparticles (QP-NP)

QP-NPs were prepared using a desolvation method, which was previously described by Ko and Gunasekaran, with slight modification [17]. Different concentrations of HSA in distilled water and tween 80 (1% *w*/*v*) were prepared, and then the pH was adjusted to 7–9 with 0.1 M NaOH as required (Table 1). QP was dissolved in the aforementioned HSA solutions at a concentration of 4 mg/mL with continuous stirring at 1000 rpm for 30 min. Drug polymer solution was titrated with absolute ethanol at a rate of 0.2 mL/min under continuous stirring at a constant stirring speed of 1000 rpm. The volume of ethanol to drug polymer solution was 1:2 *v*/*v*. Stirring was maintained for sufficient time as required (Table 1) at room temperature. The obtained NPs were harvested by centrifugation at 14,000 rpm for 30 min at 4 °C. The collected pellets were redispersed in PBS (pH 7.4) at a concentration of 1 mg/mL.

### 2.3. Characterization of the Prepared QP-NP

#### 2.3.1. Particle Size Analysis, Polydispersity Index, and Zeta Potential

Particle size, polydispersity index (PDI), and zeta potential were measured using dynamic light scattering (DLS) technique (Nanosizer ZS Series, Malvern Instruments, Southborough, MA, USA). All formulations were diluted with deionized water (1: 100 *v*/*v*) prior to measurements at 25 °C [18]. Samples (10 µg/mL) were transferred to disposable plain-folded capillary Zeta cells. All the measurements represented the average of 20 runs, and each run was completed in triplicate.

#### 2.3.2. Entrapment Efficiency (EE%) and Loading Efficiency (LE%)

Briefly, 5 mg of each prepared formulation was added to 10 mL of pH 7.4 phosphate buffer/ethanol (1:1) solution containing 1 mL of 50% (*w*/*v*) trichloroacetic acid. The system was incubated at 4 °C for 24 h. The incubated samples were centrifuged at 4000 rpm for 15 min, and the QP amount was calculated as the ratio of actual drug content to theoretical drug content [19,20]. Quantification was performed using a previously validated HPLC method described by Mandrioli et al. [21]. The EE% and LE% were calculated using the following equations:(1)$$\mathrm{EE}\%=\frac{amount\;of\;QP\;inside\;QP-NPs\;}{Total\;amount\;of\;QP\;added\;}*100$$
(2)$$\mathrm{LE}\%=\frac{mass\;of\;entrapped\;QP\;in\;QP-NPs\;}{Total\;mass\;of\;QP-NPs\;}*100$$

#### 2.3.3. Transmission Electron Microscope (TEM)

The morphological architectures of the optimized QP-NPs were visualized, as described elsewhere [22]. The QP-NP was dropped on a copper 300-mesh grid, coated with carbon. The sample was stained with phosphotungstic acid (1%), dried for 5 min, and then visualized with TEM at 200 KV.

#### 2.3.4. In Vitro Drug Release

The release of QP from QP-NPs was studied using the dialysis method [22]. Dialysis bags with a molecular weight cut-off of 10 kDa were filled with 1.5 mL of QP-NPs with equivalent drug amount of 4 mg with FCS (50% *v*/*v*). Samples were dialyzed against phosphate-buffered saline (PBS-pH; 7.4, 50 mL) containing Tween 80 (0.2% *w*/*v*). Samples were placed in thermostatically controlled shaking water bath at 250 strokes/min at 37 °C. An aliquot of 0.3 mL was withdrawn from the release medium at predetermined times, and the same volume of pre-heated fresh medium was added to maintain the final volume constant. The concentration of QP in the release media was quantified using HPLC [23].

### 2.4. In Vitro Hemolysis Assay of the Optimized QP-NPs

The hemolytic potential of the optimized QP-NPs was evaluated using rat’s red blood cells (RBCs). The RBCs were incubated with QP-NPs at different drug concentrations (0–200 µg/mL) for 2 h at 37 °C. Positive and negative controls were 0.5% *w*/*v* Triton X-100 (100% lysis) and PBS pH 7.4 (0% lysis); respectively. Samples were centrifuged at 4000 rpm for 5 min at 4 °C, and the UV absorbance of the released hemoglobin was determined at 545 nm [24].
(3)$$Hemolysis\;ratio\%=\frac{UV\;absorbance\;of\;sample-UV\;absorbance\;of\;negative\;control}{UV\;absorbance\;of\;positive\;control-UV\;absorbance\;of\;negative\;control\;}*100$$

### 2.5. In Vivo Studies of the Selected QP-NPs

#### 2.5.1. Pharmacokinetic Studies

To investigate the effectiveness of the optimized QP-NPs in improving the pharmacokinetic profiles of QP and its brain targeting ability, 198 adult male albino rats aged from 4 to 5 months and weighing about 200 g ± 10% were used. The animals were handled according to an approved protocol by the ethical committee of Al-Taif University, KSA; approval number: 44-089. Rats were distributed equally into three groups: Oral QP solution, intravenous (IV) QP solution, and IV QP-NPs at a dose of 5 mg/kg. Animals received oral QP through a gastric tube, whereas the 100 µL intravenous dose was injected to the animals through the tail vein [25]. Blood samples were withdrawn at different time intervals (*n* = 6). The samples were collected in heparinized tubes and centrifuged at 3000 rpm for 30 min at 4 °C (Laborezentrifugen, 2k15; Sigma, Osterode am Harz, Germany) to isolate the plasma samples, which were stored at −80 °C until drug analysis. At the same time intervals, the brain, lung, spleen, kidney, and liver were collected after cervical dislocation of the sacrificed animals. Organs were rinsed with ice cold normal saline in a bath sonicator (170 W and a frequency of 42 kHz) for 2 h to remove circulating blood. Consequently, organs were dried with filter paper, weighed, and stored at −80 °C until analysis. The brain and other biological tissues were homogenized with PBS pH 7.4 at 10,000 rpm using tissue homogenizer for 5 min (Thomas Scientific, McDonough, GA, USA). The tissue homogenates were centrifuged, and the supernatants were collected. The QP amount in the blood, the brain, or other tissue samples was determined by a previously described HPLC method [23].

Different Pharmacokinetic parameters, including maximum drug concentration (C_max_), the time required to reach C_max_ (T_max_), the area under the concentration−time curve (AUC_0–24_ and AUC_0–∞_), the mean residence time (MRT), and the elimination rate constant (K_el_) in the plasma and in homogenized brain tissues, were calculated using the pharmacokinetics software PKSolver Add Ins for Microsoft Excel 2007. In addition, the drug’s concentrations in different tissues (liver, kidney, spleen, and lung) were also determined at the same time intervals.

#### 2.5.2. Pharmacodynamics Studies

##### Paw Test

Paw test was performed using a Perspex platform, with dimensions of 30 × 30 × 20 cm (length × width × height). The plate contained two holes (4 cm) in the top for the forelimbs, two other holes (5 cm) in the bottom for the hindlimbs, and a slit for the tail. Twenty-four rats were randomly distributed into four groups (*n* = 6), namely, oral QP solution, IV QP solution, IV QP-NPs, and control. After 30 min of QP administration, the rats hindlimbs and forelimbs were lowered in the holes, and the tail was placed in the slit. The Forelimb Retraction Time (FRT), the time taken to withdraw one forelimb by the rat from the top holes, and the Hind limb Retraction Time (HRT), which was the time taken to withdraw one hind limb from the bottom holes, were recorded [26,27].

##### Open Field Test in Schizophrenia Rat Model

Schizophrenia-like symptoms were induced in 24 rats by a single intraperitoneal injection of ketamine (25 mg/kg) [28]. The animals were randomly divided into four groups, QP solution (oral and IV), IV QP-NPs, and an untreated group. The administered QP dose was 5 mg/kg/day for one week. The experiment was carried out on an open-field apparatus (40 × 40 × 30 cm; Accuscan Instruments, Columbus, OH) divided into 16 small squares (10 × 10 cm) by black lines, each with 18 induced schizophrenic rats placed centrally. All rats were kept in their home cages in the testing room for 2 h before being gently handled by the base of their tails and placed individually in one of the four corners of the open field facing the center. To investigate the animal’s motor activity, the total number of squares crossed by the animals over a 60 min period was recorded and compared with control untreated group (*n* = 6) [29].

### 2.6. Statistical Analysis

All in vitro experiments were performed in triplicate, and the results were recorded as the mean ± SD. Six replicates ± standard error (SE) were used to express the results of all in vivo studies. For comparing two variables, Student *t*-test was used. To compare different parameters between groups, one-way analysis of variance (ANOVA) was used, followed by the Tukey HSD test with 95% percent confidence intervals; *p* < 0.05 was deemed significant.

## 3. Results and Discussion

### 3.1. Preparation and Characterization of Quetiapine HSA Nanoparticles (QP-NPs)

To investigate the effect of different formulation factors (CPPs), including HSA concentration, medium pH, and the stirring time on the formulations’ particle sizes (Y1) and EE% (Y2), the BBD was employed. As represented in Table 2, all the prepared thirty-two formulae had a size ranging from 80.98 ± 3.12 nm to 254.87 ± 4.12 nm and a QP EE% of more than 60%. Statistical analysis (Appendix A) suggested the quadratic model as the best fit statistical model for both particle size and EE% responses based on the highest R2 (0.9583 and 0.9589) and the lowest PRESS values (8580.09 and 267.24); respectively.

Appendix A demonstrate the BBD responses’ ANOVA’s regression coefficients for particle sizes and EE% of the prepared QP-NPs at *p*-values *p* < 0.05. Additionally, all the obtained formulae had a PDI value of less than 0.25, indicating the formation of monodispersive system [30].

As represented in Equation 4 and Figure 1, particle sizes of the prepared QP-NPs were directly proportional to HSA concentrations, whereas increasing pH values or stirring time durations led to particle size reductions. Increasing HSA concentration is accompanied by higher media viscosity that opposes the influence of shear on the formation of small droplets [31,32]. The negative effect of pH on particle size had been previously reported [33]. This could be due to the effect of increased pH on surface charge, thus increasing the repulsion between the particles, preventing their aggregation and reducing the system’s particle sizes [34]. Increasing the stirring time resulted in a better distribution of the prepared NPs by an increase in the applied energy to the system, leading to particle size reductions in the prepared NPs [35].

On the contrary, the positive interaction between the medium pH and the stirring time overcomes the effects of each variable on formulae particle size (Figure 2), but to a lesser extent than either factor alone.
Particle size = +103.89 + 19.99A − 50.56B − 10.43C + 22.39BC + 51.75B^2^ + 12.52C^2^(4)

As represented in Equation (5) and Figure 3, all the investigated factors had a positive and significant effect on QP EE%. Increasing HSA concentration was generally combined with an increase in EE%. This was correlated with an increase in particle size (Figure 1A). This might have been due to the fact that the formation of larger particles increased the probability of interactions between the HSA and QP, which resulted in improving EE% [36,37]. Additionally, increasing pH and stirring time also improved QP EE% [30]. This could have been attributed to the increased contact time between the NPs and QP, which allowed for better encapsulation. On the contrary, the interaction between the increased HSA concentration and increased pH had an antagonistic effect on the QP-NPs EE%, as represented in Figure 4. This might have been due to the NPs disintegration and precipitation [38].
EE% = +91.34 + 9.79A + 3.32B + 1.57C − 2.96AB − 7.61A^2^(5)

### 3.2. Optimization of the Prepared QP-NPs

The overlay plot was contour plotted to investigate the main effects of the used factors, and the design space was constructed to optimize the tested CPPs to obtain the QP-NPs with the lowest particle sizes and the maximum EE% (Appendix A). Based on the highest desirability, one QP-NP was selected. Appendix A shows its composition and the corresponding predicted and experimental particle sizes and EE% values. The calculated error % was 4.06 and 1.4% for the particle sizes and EE%, respectively, indicating the suitability of the chosen model to predict the suitable CPPs. The LE% of the prepared selected QP-NPs formula was 7.11 ± 0.32%, with a negative zeta potential value equal to −21.35 ± 1.54 (Table 3). A zeta potential value > −20 mV indicated a high degree of system stability by increasing the inter particle repulsions, which reduced particle aggregations [39].

### 3.3. Morphological Structure of the Optimized, Selected Quetiapine-HSA- Nanoparticles

A TEM image of the selected formula is illustrated in Figure 5. QP-NPs appeared as scattered rounded nanoparticles with particle sizes (≈100 nm) consistent with the DLS technique results (Table 3).

### 3.4. In Vitro Release of QP from the Selected HSA NPs

The in vitro release profile of QP from the selected QP-NPs system was studied using the dialysis method at different time intervals for 72 h and represented in Figure 6. As demonstrated, the selected formula exhibited a biphasic release pattern, where an initial fast release was noticed within 12 h (55.86% ± 9.45%), followed by a continuous cumulative release with a consistent pattern until it reached 96.04% ± 3.19% after 72 h. These results were comparable to previous studies that demonstrated an initial burst release followed by consistency in the in vitro drug release pattern from albumin NPs [40,41]. The development of a controlled release platform would be beneficial to avoid the pre-mature release of the payload before reaching the target site.

### 3.5. The In Vitro Hemolysis Assay of the Optimized QP-NPs

To investigate the biocompatibility of the selected prepared QP-NP formula for IV injection, a hemolysis assay test was performed. Figure 7 shows that all tested concentrations had an accepted hemolytic activity (<5%) [42]. The safety of nanocarriers was the main concern for their clinical applications. Therefore, the development of a biocompatible platform as albumin NPs would be beneficial [9,10,11].

### 3.6. Pharmacokinetics Study

QP concentration in plasma and brain were quantified following IV administration of QP-NPs compared to IV and oral QP solutions (Figure 8). The pharmacokinetics parameters of the prepared IV QP-NPs in the plasma and brain were calculated to determine the ability of the selected formula in improving QP bioavailability and brain targeting (Table 4). By comparing IV QP-NPs with an IV QP solution, there was a significant increase (*p* < 0.05) in the AUC_0–24h_ [34]. Consequently, brain C_max_ and AUC_0–24h_ following IV QP-NPs showed superiority over the IV and oral administrations of the QP solution. These results indicated efficient brain deposition and increased drug absorption after the IV administration of QP-NPs (Figure 8B). In addition, the shorter brain T_max_ (2 h) compared to the oral solution (4 h) indicated an increase in the rate of drug deposition in the brain compared to oral dose (Table 4). Moreover, MRT values in both plasma and brain following IV QP-NPs were higher than the calculated values of the IV and orally administered QP solutions. In addition, the slow elimination of IV QP was indicated with the lowest K_el_. The improved brain pharmacokinetic parameters could have been attributed to the nature of HSA as an intrinsic protein and was an important biocarrier for many endogenous as well as exogenous substances [43]. Additionally, the NP size (≈100 nm) was a versatile platform for crossing the BBB [44].

The concentration of the drug in different tissues was also determined to investigate the drug biodistribution. The maximum drug concentration C_max_ in the tissues under investigation was displayed in Figure 9. The drug concentration was determined in the mentioned organs, as the liver and kidneys were the main excretion organs [45]. QP was mainly metabolized by the cytochrome CYP3A4. CYP3A4 enzymes are located in the liver, small and large intestines, testicles or ovaries, duodenum, pancreas, kidneys, spleen, and lymph nodes [46]. The heart and the kidneys are the main known sites of major QP side effects [47,48,49]. As represented in Figure 9, there was a significant increase in the QP in the rat brain and spleen after IV QP-NPs over the IV or oral QP solutions. The highest drug concentration was noticed in the brain, which was the target organ. The results of the tissue distribution study confirmed the ability of the IV QP-NPs to improve the brain targeting and, subsequently, the therapeutic efficacy, with relatively lower adverse effects compared to the IV QP solution. In addition, the spleen as an organ of the reticuloendothelial system is a major fate for different types of NPs [30].

### 3.7. Pharmacodynamics Studies

The antipsychotic effects of the selected IV QP-NP, IV, and oral QP solutions were studied using the paw and the open field tests [50].

#### 3.7.1. Paw Test

The paw test was used to study locomotor activity of the tested rats as an indication of the drug’s antipsychotic activity [51]. The potential antipsychotic effect is indicated by increasing the hindlimb retraction time (HRT), and the possibility for extrapyramidal side effects is associated with an increase in the forelimb retraction time (FRT) [51]. In this study, and as represented in Figure 10A, the IV administration of the QP-NPs showed significantly higher HRT and lower FRT values when compared to the IV and oral QP solutions. QP had a strong affinity for the 5-HT2 receptors, which, upon blocking, were responsible for the lack of effect on the FRT [52,53]. By inspecting Table 4, it could be noticed that the IV QP-NPs had higher AUC_0–∞_ and MRT values than the IV QP solution in both the plasma and brain. In addition, a lower K_el_ was observed in the plasma and brain following the IV administration of the QP-NPs compared to the drug solution. These results could have been correlated with the improved antipsychotic activity of the proposed system in terms of significantly higher HRT (*p* < 0.05). Therefore, the improved HRT result could have been attributed to the preferential drug deposition in brain following the IV administration of QP-NPs. On the contrary, the significant reduction in FRT in rats following the administration of IV QP-NPs compared to the IV and oral solutions was an indication for the reduction in parkinsonian symptoms accompanying QP administration (*p* < 0.05) [54]. The QP’s adverse effects could have been attributed to its off-target distribution in the body’s organs. Moreover, the data illustrated in Figure 9 revealed a significantly higher distribution of QP in the liver, lung, heart, and kidneys following the IV solution administration over the NPs (*p* < 0.05). These results might have led to significantly higher FRT values and consequently expected adverse effects.

#### 3.7.2. Open Field Test

Untreated schizophrenic rats exhibited hyperlocomotion activity, as evidenced by a significant increase in the number of crossed squares when compared to animals receiving QP in different doses or healthy rats (*p* < 0.001). The reduction in the locomotor activities of the schizophrenic-induced rats was a sign of the functional alterations of the motor neurons’ and/or GABAergic interneurons’ effects of the QP treatment, which were marked by the number of crossed squares in 60 min [55,56]. As represented in Figure 10B, rats treated by the QP-NPs significantly crossed the smallest number of squares compared to the rats who received a QP solution either orally or via an IV (*p* < 0.05). The observed reduction in the locomotion could have been attributed to the antagonistic effect of quetiapine on the 5-HT2 receptor [57]. The latter resulted in a spontaneous decrease in the ketamine-induced stimulation effect. Therefore, the IV administration of QPNPs resulted in improved brain delivery in a rat model of schizophrenia and consequently triggered the therapeutic outcome.

## 4. Conclusions

In this study, brain-targeting IV QP-NPs formulations were prepared based on identifying the main formulation factors that could affect the albumin QP-NPs’ characteristics during preparation, using a desolvation technique. These studied factors were, namely, HSA concentration, pH, and stirring time. The main aim was to prepare QP-NPs with maximum EE% values and minimum particle sizes (<100 nm) to achieve efficient brain depositions and reductions in the QP extrapyramidal side effects compared to the IV drug solution. The selected negatively charged biocompatible QP-NPs had spherical shapes, with particle sizes of 103.54 ± 2.36 nm and EE% = 96.32 ± 3.98. The IV QP-NPs had a significant increase in the rate and extent of the QP absorption over the IV and oral drug solutions by 1.41- and 4.84-fold, respectively. Pharmacodynamic results proved the ability of IV QP-NPs to trigger therapeutic efficiency in schizophrenic rats compared to the IV and oral QP solutions. These results provided a rationale for further toxicological and clinical studies.

## Figures and Tables

**Figure 1 pharmaceutics-15-01785-f001:**
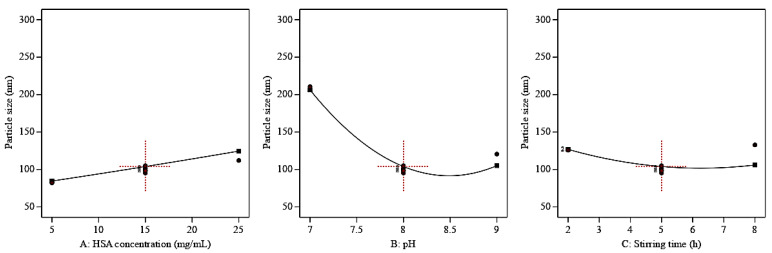
Plots of the main effect of different significant critical process parameters on particle size (Y1). Increasing HSA concentration (**A**) had a positive influence on particle size while a negative effect was seen by increasing pH (**B**) and stirring time (**C**).

**Figure 2 pharmaceutics-15-01785-f002:**
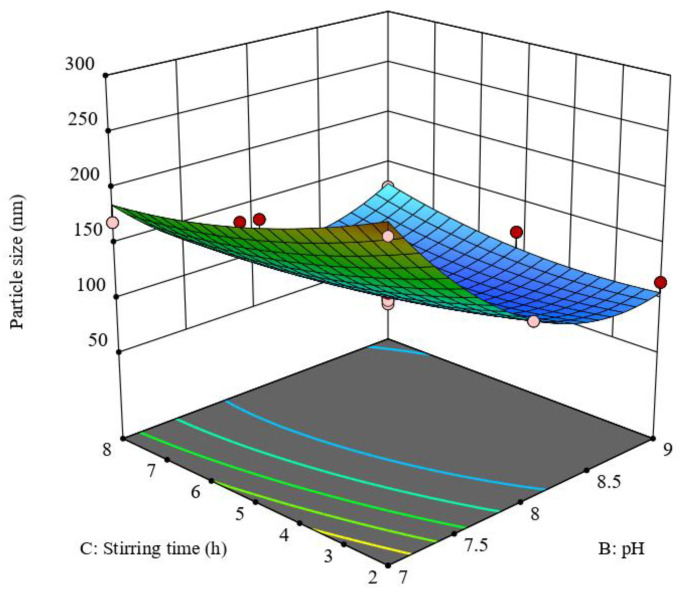
Response 3D plot for the significant parameters interaction on QP-NPs particle size (Y1). The positive interaction between pH and stirring time indicates a direct relation between the interaction of both variables and QP-NPs particle sizes.

**Figure 3 pharmaceutics-15-01785-f003:**
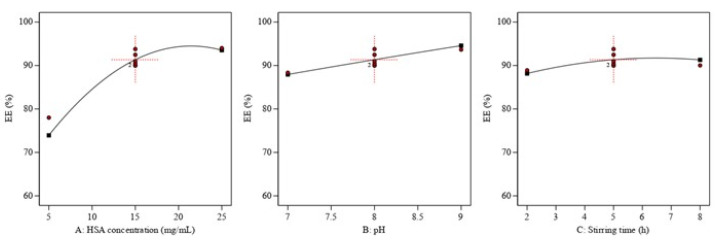
Plots of the main effect of different significant critical process parameters on EE% (Y2). (**A**): HSA concentration, (**B**): pH, and (**C**): stirring time had a positive influence on EE%.

**Figure 4 pharmaceutics-15-01785-f004:**
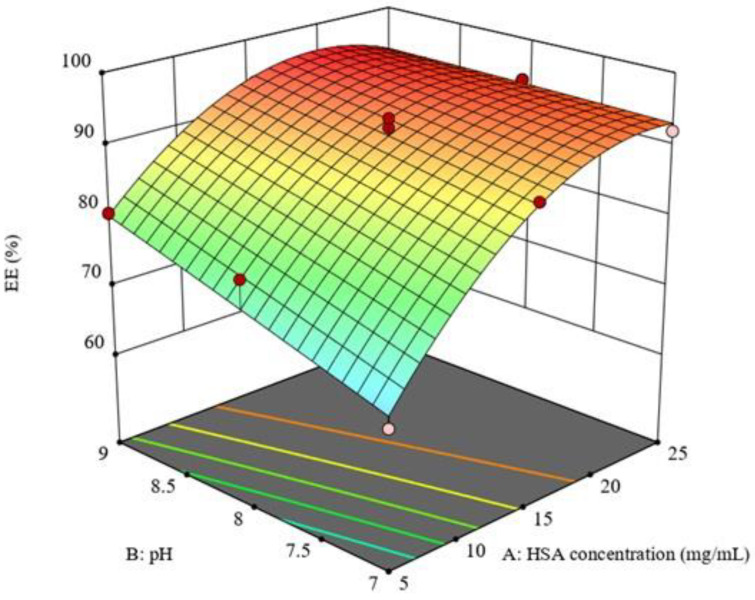
Response 3D plot for the interaction of (AB) between HSA concentration and pH on EE% (Y2). An inverse relationship was depicted for the interaction between both factors and QP-NPs EE%.

**Figure 5 pharmaceutics-15-01785-f005:**
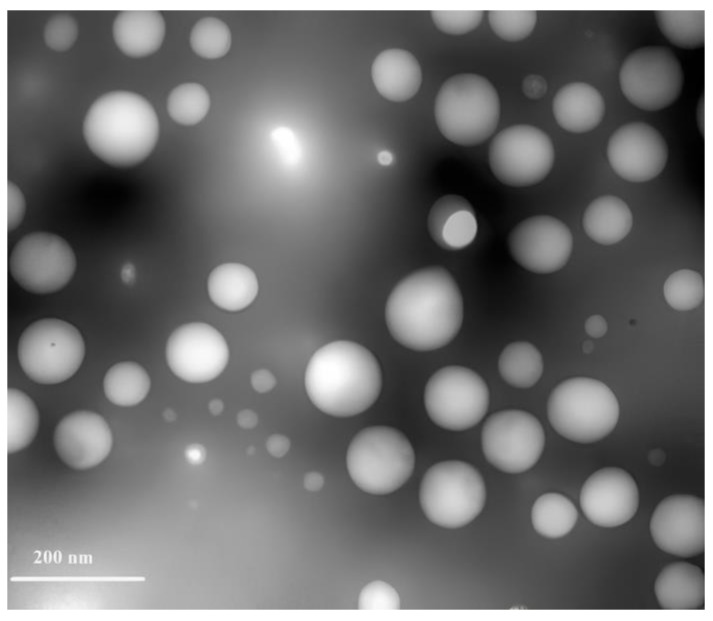
Morphological characterization of the optimized QP-NPs by transmission electron microscope. QP-NPs appeared as scattered rounded shaped nanostructures with particle sizes consistent with DLS technique.

**Figure 6 pharmaceutics-15-01785-f006:**
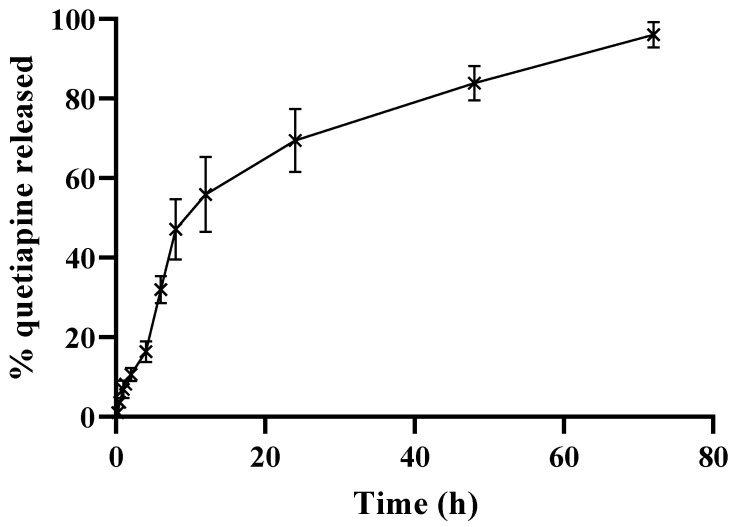
In vitro release profile of drug from QP-NPs. In vitro quetiapine release from the optimized albumin NPs in PBS (pH 7.4) in presence of FCS (50% *v*/*v*). Drug concentration in the dialysate was quantified by HPLC. Data point represents mean and SD (*n* = 3).

**Figure 7 pharmaceutics-15-01785-f007:**
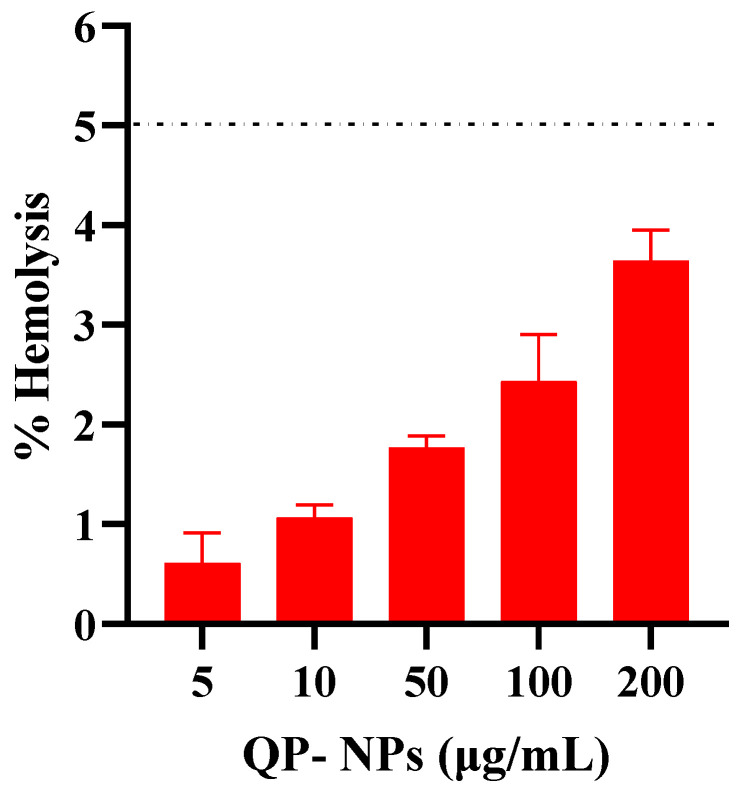
The in vitro hemolysis assay of the optimized quetiapine albumin NPs. Rat RBCs were incubated with QP-NPs at different drug concentrations (0–200 µg/mL) for 2 h at 37 °C. Positive and negative controls were 0.5 *w*/*v*% Triton X-100 and PBS (pH 7.4), respectively. Samples were centrifuged at 4000 rpm for 5 min at 4 °C and the absorbance of the released hemoglobin was determined at 545 nm. Data point represents mean and SD (*n* = 3). The dotted line represents the acceptable hemolysis range.

**Figure 8 pharmaceutics-15-01785-f008:**
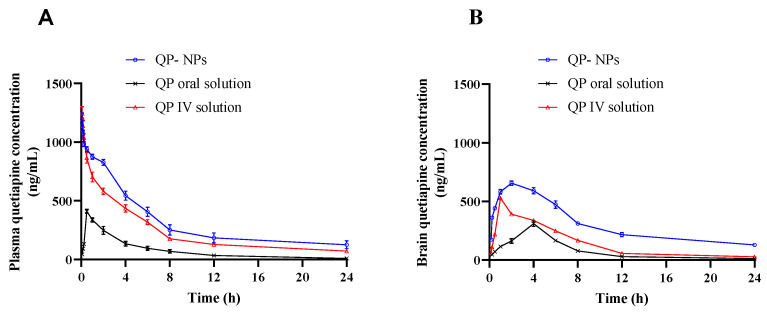
Quetiapine concentrations in rats’ plasma (**A**) and brain (**B**) after administration of various formulations. Animals received a dose of 5 mg/kg of QP either via IV injection through the tail vein or oral solution. At each time point, six animals were sacrificed from each group, and the concentration of QP in plasma and brain was quantified using HPLC. A significantly higher brain QP concentration was observed at all time points following intravenous administration of QP-NPs compared to IV solution or oral solution. Datapoints represent the mean ± SE (*n* = 6).

**Figure 9 pharmaceutics-15-01785-f009:**
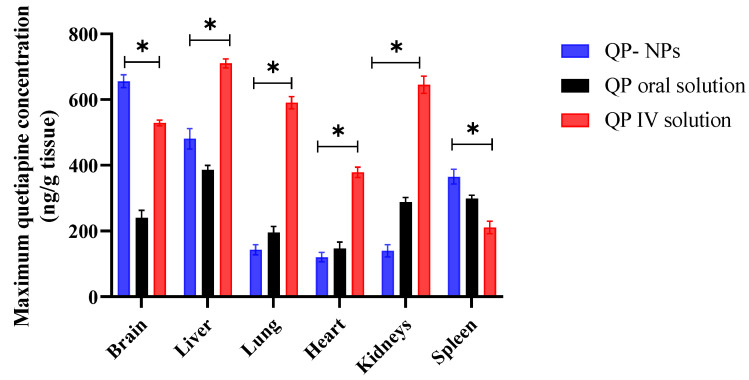
Maximum Quetiapine concentration in spleen, brain, lung, liver, and kidneys following IV QP-NPs, oral and IV QP administration to rats. IV QP HSA NPs showed the highest deposition in brain and spleen. On the contrary, IV QP solution had the highest distribution in liver, lung, heart, and kidneys. Each value represents the mean and standard error (*n* = 6). Statistical analysis was conducted using ANOVA followed by the Tukey HSD test at * *p* < 0.05.

**Figure 10 pharmaceutics-15-01785-f010:**
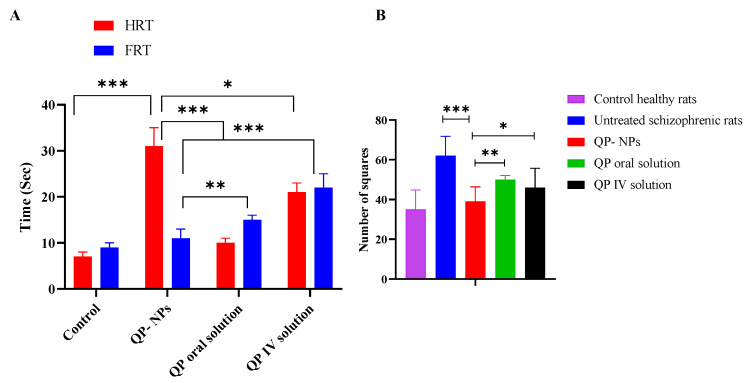
Assessment of pharmacodynamics effect of IV QP-NPs in (**A**) Paw test (**B**) ketamine induced schizophrenia in rats using open field test. Each value represents the mean and standard error (*n* = 6). Statistical analysis was conducted using ANOVA followed by the Tukey HSD test at * *p* < 0.05, ** *p* < 0.01, *** *p* < 0.001.

**Table 1 pharmaceutics-15-01785-t001:** Critical process parameters levels, quality attributes, and quality target product profiles for the preparation of QP-NPs using the BBD.

Critical Process Parameters (Coded Independent Variables)	Levels		
	Low (−1)	Medium (0)	High (1)
A: HSA concentration (mg/mL)	5	15	25
B: pH	7	8	9
C: Stirring time (h)	2	5	8
Critical Quality attributes (Responses)	Quality target product profile (Constrains)		
Y1: Particle size (nm)	Minimum		
Y2: Entrapment efficiency EE (%)	Maximize		

**Table 2 pharmaceutics-15-01785-t002:** Experimental design matrix of the CPPs and the related CQAs.

Run	Critical Process Parameters (CPPs)			Critical Quality Attributes (CQAs)	
	A: HSA Concentration (mg/mL)	B: pH	C: Stirring Time (h)	Particle Size (nm) ^a,c^	EE% ^b,c^
1	5	7	8	153.2 ± 3.25	70.12 ± 1.25
2	15	8	5	98.5 ± 2.5	90.05 ± 2.35
3	5	8	8	82.41 ± 3.21	72.55 ± 1.45
4	15	8	5	98.3 ± 1.25	91 ± 2.65
5	25	7	2	274 ± 3.47	89.78 ± 3.45
6	15	8	5	101.74 ± 2.65	92.47 ± 2.47
7	15	8	5	95.3 ± 4.98	93.78 ± 2.36
8	15	9	2	115.2 ± 5.11	90.06 ± 2.14
9	25	7	8	205.4 ± 6.78	90.78 ± 3.45
10	5	7	2	230.7 ± 8.45	60.78 ± 2.54
11	5	8	2	105.41 ± 3.21	67.71 ± 1.45
12	15	8	8	132.78 ± 2.15	90.03 ± 2.65
13	25	8	2	151 ± 1.36	88.73 ± 3.56
14	15	9	8	125.94 ± 2.51	91.45 ± 2.78
15	15	8	5	100.5 ± 2.41	89.97 ± 3.45
16	15	8	5	105 ± 2.54	90.45 ± 2.45
17	15	9	5	120.5 ± 2.36	93.65 ± 2.14
18	25	8	5	112 ± 3.14	94 ± 3.24
19	5	7	5	196 ± 2.45	65.89 ± 2.45
20	25	8	8	142.65 ± 3.14	95.78 ± 1.47
21	15	7	2	240 ± 2.45	87.65 ± 3.24
22	25	9	2	132.87 ± 2.54	94.78 ± 2.54
23	15	7	5	210.6 ± 2.35	88.34 ± 2.45
24	15	8	2	125.78 ± 3.14	88.9 ± 3.24
25	25	9	5	97.8 ± 2.45	92.47 ± 2.87
26	15	7	8	170 ± 2.14	90.79 ± 3.47
27	5	9	8	120 ± 1.45	80.79 ± 3.15
28	25	9	8	135.69 ± 3.65	93.23 ± 2.84
29	25	7	5	254.87 ± 4.12	92 ± 3.78
30	5	8	5	81.98 ± 3.25	78 ± 4.15
31	5	9	5	95.68 ± 2.45	80.48 ± 1.25
32	5	9	2	80.98 ± 3.12	78.94 ± 3.12

^a^ Particle size was measured by DLS. ^b^ Calculated as percentage of initial QP added, determined directly by HPLC. ^c^ Expressed as mean ± SD (*n* = 3).

**Table 3 pharmaceutics-15-01785-t003:** The in vitro characterization of the optimized QP-NPs.

HSA Concentration (mg/mL)	pH	Stirring Time (h)	Particle Size (nm) ^a,e^	PDI ^a,c^	Zeta Potential (mV) ^b,e^	EE% ^c,e^	LE % ^d,e^
18	8.6	5	103.54 ± 2.36	0.152 ± 0.005	−21.35 ± 1.54	96.32 ± 3.98	7.11 ± 0.32

^a^ Particle size was measured by DLS. ^b^ Zeta potential determined by electrophoresis. ^c^ Calculated as percentage of initial quetiapine added, determined directly by HPLC. ^d^ Calculated as percentage of entrapped quetiapine weight to the total albumin NPs weight. ^e^ Expressed as mean ± SD (*n* = 3).

**Table 4 pharmaceutics-15-01785-t004:** Pharmacokinetic parameters of intravenous Quetiapine solution, oral solution, and intravenous QP-NPs.

Parameter	Plasma			Brain		
	QP-NPs	IV QP Solution	Oral QP Solution	QP-NPs	IV QP Solution	Oral QP Solution
C_max_ (ng/mL)	-	-	410.62 ± 17.51	655.93 ± 19.53	528.83 ± 8.77	310.62 ± 23.67
T_max_ (h)	-	-	0.5	2	1	4
AUC_0–24h_ (µg/mL.h)	7.48	5.58	1.81	7.24	1.87	1.72
AUC_0–∞_ (µg/mL.h)	9.05	6.36	1.87	8.94	1.96	1.82
MRT (h)	10.15	8.71	5.97	11.25	7.12	7.4
K_el_ (h^−1^)	0.08	0.09	0.14	0.076	0.14	0.13
Absolute bioavailability %	141.5	100	29.4	-	-	-
Relative bioavailability %	483.95	340.11	-	-	-	-

## Data Availability

Data will be made available on request.

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
