# Peer review of "Quetiapine Albumin Nanoparticles as an Efficacious Platform for Brain Deposition and Potentially Improved Antipsychotic Activity"

_pharmaceutics, 2023, doi:10.3390/pharmaceutics15071785_

Round 1

Reviewer 1 Report

This is a well-designed manuscript. However, I have a few minor comments.

1. Please provide a detailed description of the methods. For example, the nanoparticle preparation method provides the volume of ethanol used as well the speed of addition.

2. Have you normalized the brain drug concentration for the presence of blood capillaries?

3. The authors only explain the results. However, the significance of the results is not discussed in detail.

4. Minor comments:

Line 312: Please remove the comment "how much".

The quality of English is fine.

Author Response

Comments and Suggestions for Authors

This is a well-designed manuscript. However, I have a few minor comments.

  1. Please provide a detailed description of the methods. For example, the nanoparticle preparation method provides the volume of ethanol used as well the speed of addition.

The following details were added to the following experiments as follows:

2.2.2. Preparation of Quetiapine-loaded human serum albumin nanoparticles (QP-NP)

Drug polymer solution was titrated with absolute ethanol at rate 0.2 mL/ min under continuous stirring at a constant stirring speed of 1000 rpm. The volume of ethanol to drug polymer solution was 1:2 v/v. Stirring was maintained for sufficient time as required (Table 1) at room temperature. The obtained NPs were harvested by centrifugation at 14,000 rpm for 30 min at 4 °C. The collected pellets were redispersed in PBS (pH 7.4) at a concentration of 1 mg/mL.

2.3.1. Particle size analysis, polydispersity index and zeta potential

Particle size, polydispersity index (PDI) and zeta potential were measured using dynamic light scattering (DLS) technique (Nanosizer ZS Series, Malvern Instruments, Southborough, MA). All formulations were diluted with deionized water (1: 100 v/v) prior to measurements at 25 °C [18]. Samples (10 µg/mL) were transferred to disposable plain folded capillary Zeta cells. All the measurements represent the average of 20 runs, each run was completed in triplicate.

2.3.3. Transmission electron microscope (TEM)

The morphological architecture of the optimized QP - NPs was visualized as described elsewhere [22]. The QP-NP was dropped on a copper 300-mesh grid, coated with carbon. The sample was stained with phosphotungstic acid (1%) and dried for 5 min then visualized with TEM at 200 KV.

2.5.2.1. Paw test

Paw test was performed using a Perspex platform with dimensions of 30* 30* 20 cm (length*width*height). The plate contained two holes (4 cm) in the top for the fore-limbs, two other holes (5 cm) in the bottom for the hindlimbs and a slit for the tail. Twenty-four rats were randomly distributed into four groups (n=6) namely, oral QP solution, IV QP solution, IV QP -NPs and control. After 30 min of QP administration, the rats hindlimbs and forelimbs were lowered in the holes and the tail is placed in the slit. The Forelimb Retraction Time (FRT); the time taken to withdraw one forelimb by the rat from the top holes and the Hind limb Retraction Time (HRT); which the time taken to withdraw one hind limb from the bottom holes were recorded [26, 27].

  1. Have you normalized the brain drug concentration for the presence of blood capillaries?

The authors would like to thank the reviewer for his valuable comment. We agree with the reviewer that the brain drug concentration should be normalized for the presence of blood capillaries. Therefore, organs were rinsed with ice cold saline for 2 h in bath sonicator. Therefore, the following sentence was added to the manuscript as follows:

Organs were rinsed with ice cold normal saline in bath sonicator (170 W and a frequency of 42 kHz) for 2 h to remove circulating blood. Consequently, organs were dried with filter paper, weighed, and stored at −80 °C until analysis.

  1. The authors only explain the results. However, the significance of the results is not discussed in detail.

More details about the significance of the data were added to the results and discussion in the revised manuscript.

 Minor comments:

Line 312: Please remove the comment "how much"

The authors apologize for this typo-error and the comment was deleted.

Reviewer 2 Report

The paper describes a reproducible formulation of quietiapine in HSA. The complex can pass the BBB in rats. The work looks nice and is well-documented.

Author Response

Comments and Suggestions for Authors

The paper describes a reproducible formulation of quetiapine in HSA. The complex can pass the BBB in rats. The work looks nice and is well-documented.

The authors are very grateful to the reviewers’ meticulous revision and comment.

Reviewer 3 Report

In this work, Abdel-Bar et al. present a novel nanomaterial comprising an antipsychotic drug in albumin nanoparticles for drug delivery of the drug. The authors describe the synthesis and the physicochemical properties of the material, and test the pharmacokinetics of the release as well as the biodistribution and bioavailability of the drug in vivo, after intravenous administration. The authors demonstrate an improved antipsychotic activity in rats by behavioral testing, claiming improved drug brain targeting and antipsychotic efficiency.

In general, this is a well-written and comprehensive manuscript with interesting results, although the research presented here is not excessively novel. Albumin nanoparticles for enhanced drug delivery to the brain have been extensively described in literature. This is another drug for another disease, but the degree of novelty is somehow limited.

Some issues need discussion from the authors:

- It is not clear why minimum nanoparticle size is a target for the design, as nanoparticles ranging 50-150 nm are useful for in vivo applications, and a very low size may favor renal clearance of the particles. The authors should clarify what they understand by “minimum” size and why this is an advantage over medium sized nanoparticles.

- Page 3. Line 102: “All samples were diluted with deionized water (1: 100 v/v)” this sentence tells nothing, if the initial concentration is not defined. Please give an approximate value (circa…) or range for the concentration of particles in w/v (mg or mol / ml or similar).

- Page 4, line 149. Organs were rinsed in saline, but I understand that rats were nor transcardially perfused to remove circulating blood. How long where the organs rinsed and was mechanical energy applied to ensure a good clearance from circulating blood from the organs?.

- Page 5, line 166: in the Forelimb Retraction Time tests, the authors mention that this is the time taken by the animal to withdrawn the par. The question is withdrawn from where? or from what?. Cand you give more details about the test?.

- page 5, lines 174-175: The authors claim that “Schizophrenia was induced in 24 rats by intraperitoneal injection of ketamine (25 mg/kg) [28]”. The authors must clarify if they have used a single Ketamine dose or a sub-chronic administration (indicating timing) since the symptoms are different depending on the administration type. On the other hand this is a model of disease inducing schizophrenia-like symptoms but not necessarily inducing the disease in the animals. Please rephrase the sentence.

- page 12, line  312: “the AUC0-24h (how much?)[34].” Please correct typo

- page 12, lines 323-336. The whole paragraph is repeated in lines 354-367

- Figure 10, the explanation provided by the authors respect to the high difference in HRT and FRT in nanoparticle treated animals is not totally convincing to me. T. Specially because this effect (the large different between the two times) is not observed in iv treated animals. Please elaborate better this section.

Author Response

In this work, Abdel-Bar et al. present a novel nanomaterial comprising an antipsychotic drug in albumin nanoparticles for drug delivery of the drug. The authors describe the synthesis and the physicochemical properties of the material, and test the pharmacokinetics of the release as well as the biodistribution and bioavailability of the drug in vivo, after intravenous administration. The authors demonstrate an improved antipsychotic activity in rats by behavioral testing, claiming improved drug brain targeting and antipsychotic efficiency.

In general, this is a well-written and comprehensive manuscript with interesting results, although the research presented here is not excessively novel. Albumin nanoparticles for enhanced drug delivery to the brain have been extensively described in literature. This is another drug for another disease, but the degree of novelty is somehow limited.

Some issues need discussion from the authors:

- It is not clear why minimum nanoparticle size is a target for the design, as nanoparticles ranging 50-150 nm are useful for in vivo applications, and a very low size may favor renal clearance of the particles. The authors should clarify what they understand by “minimum” size and why this is an advantage over medium sized nanoparticles.

The authors would like to thank the reviewer for raising such valuable comment. We agree with the reviewer that particle size is one of the most important factors that could affect the fate and biodistribution of the administrated nanoparticles. Several reports revealed that nanoparticles with particle size around 10 nm are rapidly eliminated by kidneys [1]. In addition, nanoparticles with size more than 200 nm showed rapid accumulation in liver and spleen [2]. Gao and Jiang observed an inverse relation between methotrexate-loaded polybutylcyanoacrylate nanoparticles size and brain deposition where the highest drug accumulation in brain was observed in nanoparticles with size ≈ 70 nm [3]. In addition, Cruz and coworkers studied the effect of poly(lactic-co-glycolic) nanoparticles size in the range from 100 to 800 nm on brain deposition where the highest brain deposition was obtained with the smallest particles (100 nm) [4]. The particle size obtained in this study was in the range of 80.98 ± 3.12 nm to 254.87 ± 4.12 nm. Therefore, the minimum particle size range will be less likely to be rapidly cleared by kidneys. In addition, the optimized formulae in this study has a respective predicted and experimental particle size of  97.48 and 103.54±2.36 nm which is close to the particle size range that previously showed preferential brain deposition.

References

  1. Zuckerman JE, Choi CHJ, Han H, Davis ME. Polycation-siRNA nanoparticles can disassemble at the kidney glomerular basement membrane. Proc. Natl Acad. Sci. USA. 2012;109:3137–3142.
  2. Kulkarni SA, Feng SS. Effects of particle size and surface modification on cellular uptake and biodistribution of polymeric nanoparticles for drug delivery. Pharmaceut. Res. 2013;30:2512–2522.
  3. Gao K, Jiang X. Influence of particle size on transport of methotrexate across blood brain barrier by polysorbate 80-coated polybutylcyanoacrylate nanoparticles. Int. J. Pharm. 2006, 310, 213–219.
  4. Cruz LJ., Stammes MA, Que I, van Beek ER, Knol-Blankevoort V., Snoeks TJA, Chan A, Kaijzel E.L, Löwik CWGM. Effect of PLGA NP size on efficiency to target traumatic brain injury. J. Control. Release 2016, 223, 31–41.

- Page 3. Line 102: “All samples were diluted with deionized water (1: 100 v/v)” this sentence tells nothing, if the initial concentration is not defined. Please give an approximate value (circa…) or range for the concentration of particles in w/v (mg or mol / ml or similar).

According to the reviewer valuable comment, the following details have been added to particle size analysis where the final concentration of nanoparticles was 10 µg/mL.

2.3.1. Particle size analysis, polydispersity index and zeta potential

Particle size, polydispersity index (PDI) and zeta potential were measured using dynamic light scattering (DLS) technique (Nanosizer ZS Series, Malvern Instruments, Southborough, MA). All formulations were diluted with deionized water (1: 100 v/v) prior to measurements at 25 °C [18]. Samples (10 µg/mL) were transferred to disposable plain folded capillary Zeta cells. All the measurements represent the average of 20 runs, each run was completed in triplicate.

- Page 4, line 149. Organs were rinsed in saline, but I understand that rats were nor transcardially perfused to remove circulating blood. How long where the organs rinsed and was mechanical energy applied to ensure a good clearance from circulating blood from the organs?.

The authors would like to thank the reviewer for his valuable comment. We agree with the reviewer that the rats were not transcardially perfused to remove circulating blood. Instead, organs were rinsed with ice cold saline for 2 h in bath sonicator. Therefore, the following sentence was added to the manuscript as follows:

Organs were rinsed with ice cold normal saline in bath sonicator (170 W and a frequency of 42 kHz) for 2 h to remove circulating blood. Consequently, organs were dried with filter paper, weighed, and stored at −80 °C until analysis.

- Page 5, line 166: in the Forelimb Retraction Time tests, the authors mention that this is the time taken by the animal to withdrawn the par. The question is withdrawn from where? or from what?. Cand you give more details about the test?.

Paw test is mentioned in detail as follow

Paw test was performed using a Perspex platform with dimensions of 30* 30* 20 cm (length*width*height). The plate contained two holes (4 cm) in the top for the forelimbs, two other holes (5 cm) in the bottom for the hindlimbs and a slit for the tail. Twenty-four rats were randomly distributed into four groups (n=6) namely, oral QP solution, IV QP solution, IV QP -NPs and control. After 30 min of QP administration, the rats hindlimbs and forelimbs were lowered in the holes and the tail is placed in the slit. The Forelimb Retraction Time (FRT); the time taken to withdraw one forelimb by the rat from the top holes and the Hind limb Retraction Time (HRT); which the time taken to withdraw one hind limb from the bottom holes were recorded [26, 27].

- page 5, lines 174-175: The authors claim that “Schizophrenia was induced in 24 rats by intraperitoneal injection of ketamine (25 mg/kg) [28]”. The authors must clarify if they have used a single Ketamine dose or a sub-chronic administration (indicating timing) since the symptoms are different depending on the administration type. On the other hand this is a model of disease inducing schizophrenia-like symptoms but not necessarily inducing the disease in the animals. Please rephrase the sentence.

The authors thank the reviewer for this valuable comment. The schizophrenia-like symptoms were induced using a single intraperitoneal injection of ketamine (25 mg/kg). Therefore, the sentence was rephrased as follow:

Schizophrenia-like symptoms were induced in 24 rats by a single intraperitoneal injection of ketamine (25 mg/kg) [28].

- page 12, line 312: “the AUC0-24h (how much?)[34].” Please correct typo

The authors apologize for this typo-error and the comment was deleted.

- page 12, lines 323-336. The whole paragraph is repeated in lines 354-367

 The authors apologize for this mistake and the repeated paragraph from lines 354 to 367 was deleted.

- Figure 10, the explanation provided by the authors respect to the high difference in HRT and FRT in nanoparticle treated animals is not totally convincing to me. T. Specially because this effect (the large different between the two times) is not observed in iv treated animals. Please elaborate better this section.

The antipsychotic potential effects of antipsychotic drugs are linked to an increase in HRT, while the possibility for extrapyramidal side effects was associated with a rise in FRT. Indeed, QP adverse effects are due to its off-target distribution in the body. By inspecting table 4, it could be noticed that QP-NPs has higher AUC0-∞, mean residence time (MRT) than IV QP solution in both plasma and brain. In addition, a lower Kel observed in plasma and brain following IV administration of QP-NPs compared to drug solution. These results could be correlated with the improved antipsychotic activity of the proposed system in terms of significantly higher HRT (p< 0.05). Moreover, the data illustrated in figure 9 reveled a significantly higher distribution of QP in liver, lung, heart and kidneys following IV solution administration over NPs (p< 0.05). These results might lead to the significantly higher FRT and consequently expected adverse effects.

The following sentences were added to the revised manuscript as follows:

“By inspecting Table 4, it could be noticed that IV QP-NPs has higher AUC0-∞ and MRT than IV QP solution in both plasma and brain. In addition, a lower Kel was observed in plasma and brain following IV administration of QP-NPs compared to drug solution. These results could be correlated with the improved antipsychotic activity of the proposed system in terms of significantly higher HRT (p< 0.05).

The QP adverse effects could be attributed to its off-target distribution in the body organs. Moreover, the data illustrated in Figure 9 reveled a significantly higher distribution of QP in liver, lung, heart and kidneys following IV solution administration over NPs (p< 0.05). These results might lead to the significantly higher FRT and consequently expected adverse effects.

Reviewer 4 Report

The topic of the manuscript is very interesting and timely. Finding an albumin-based drug nanocarrier holds great promise for other diseases. Drug transport based on nanoplatforms not only improves the effect of drugs but also allows the use of targeted therapies. Improving the treatment of patients with mental health problems is necessary for a better quality of life for both patients and their carers.

Let me make a few comments on the manuscript:

Verses 323 to 336 and verses 354 to 367 are practically the same. They differ in single words, e.g.

„The heart and the kidneys are the main known sites of major QP side effects [47-49]. As represented in Figure 9, there is a significant increase in the QP in the rat brain and spleen after IV QP-NPs over IV or oral QP solution. The highest drug concentration was noticed in the brain which is the target organ. The results of tissue distri- bution study confirmed the ability of IV QP-NPs to improve the brain targeting and subsequently, therapeutic efficacy with relatively lower adverse effects compared to IV QP solution. In addition, spleen as an organ of the reticuloendothelial system is a major fate for different types of NPs [30].

Besides, heart and kidney are the main known sites of major QP side effects [47-49]. As represented in Figure 9, there is a significant increase in the QP in the rat brain and spleen after IV QP NPs over IV or oral QP solution. The highest drug concentration was noticed in the brain which is the targeting organ. The results of tissue distribution study confirmed the ability of IV QP albumin NPs to improve the brain targeting and subsequently, therapeutic efficacy with relatively lower adverse effects compared to IV QP solution. In addition, spleen as an organ of the reticuloendothelial system is a major fate for different types of NPs [30].

I don't think it was intended.

There is a comma in line 223, or rather it shouldn't be there.

„Particle size= +103.89+ 19.99A- 50.56B, -10.43C+ 22.39BC+ 51.75B2+ 12.52C2 Equation 4”

Author Response

Comments and Suggestions for Authors

The topic of the manuscript is very interesting and timely. Finding an albumin-based drug nanocarrier holds great promise for other diseases. Drug transport based on nanoplatforms not only improves the effect of drugs but also allows the use of targeted therapies. Improving the treatment of patients with mental health problems is necessary for a better quality of life for both patients and their carers.

Let me make a few comments on the manuscript:

  • Verses 323 to 336 and verses 354 to 367 are practically the same.They differ in single words, e.g.

 „The heart and the kidneys are the main known sites of major QP side effects [47-49]. As represented in Figure 9, there is a significant increase in the QP in the rat brain and spleen after IV QP-NPs over IV or oral QP solution. The highest drug concentration was noticed in the brain which is the target organ. The results of tissue distri- bution study confirmed the ability of IV QP-NPs to improve the brain targeting and subsequently, therapeutic efficacy with relatively lower adverse effects compared to IV QP solution. In addition, spleen as an organ of the reticuloendothelial system is a major fate for different types of NPs [30].

 Besides, heart and kidney are the main known sites of major QP side effects [47-49]. As represented in Figure 9, there is a significant increase in the QP in the rat brain and spleen after IV QP NPs over IV or oral QP solution. The highest drug concentration was noticed in the brain which is the targeting organ. The results of tissue distribution study confirmed the ability of IV QP albumin NPs to improve the brain targeting and subsequently, therapeutic efficacy with relatively lower adverse effects compared to IV QP solution. In addition, spleen as an organ of the reticuloendothelial system is a major fate for different types of NPs [30].”

I don't think it was intended.

 The authors apologize for this mistake and the repeated paragraph from lines 354 to 367 was deleted.

  • There is a comma in line 223, or rather it shouldn't be there.

„Particle size= +103.89+ 19.99A- 50.56B, -10.43C+ 22.39BC+ 51.75B2+ 12.52C2 Equation 4”

The comma was deleted.

Reviewer 5 Report

From my point of view the manuscript entitled "Quetiapine albumin nanoparticles as an efficacious platform for brain deposition and potentially improved antipsychotic activity" is an interesting work that are very well written. The manuscript is well structured and organized. The cited papers are relevant and the discussion in each section is concise and to the point. I would highly recommend it to be published in Pharmaceutics Journal.

Author Response

Comments and Suggestions for Authors

From my point of view the manuscript entitled "Quetiapine albumin nanoparticles as an efficacious platform for brain deposition and potentially improved antipsychotic activity" is an interesting work that are very well written. The manuscript is well structured and organized. The cited papers are relevant and the discussion in each section is concise and to the point. I would highly recommend it to be published in Pharmaceutics Journal.

The authors appreciate the valuable reviewer revision and comments in the manuscript.
